# SCALABLE BAYESIAN INVERSE REINFORCEMENT LEARNING

**Alex J. Chan**
University of Cambridge, Cambridge, UK
`alexjchan@maths.cam.ac.uk`

**Mihaela van der Schaar**
University of Cambridge, Cambridge, UK
University of California, Los Angeles, USA
Cambridge Centre for AI in Medicine, UK
The Alan Turing Institute, London, UK
`mv472@cam.ac.uk`

## ABSTRACT

Bayesian inference over the reward presents an ideal solution to the ill-posed nature of the inverse reinforcement learning problem. Unfortunately current methods generally do not scale well beyond the small tabular setting due to the need for an inner-loop MDP solver, and even non-Bayesian methods that do themselves scale often require extensive interaction with the environment to perform well, being inappropriate for high stakes or costly applications such as healthcare. In this paper we introduce our method, *Approximate Variational Reward Imitation Learning* (AVRIL), that addresses both of these issues by jointly learning an approximate posterior distribution over the reward that scales to arbitrarily complicated state spaces alongside an appropriate policy in a completely offline manner through a variational approach to said latent reward. Applying our method to real medical data alongside classic control simulations, we demonstrate Bayesian reward inference in environments beyond the scope of current methods, as well as task performance competitive with focused offline imitation learning algorithms.

## 1 INTRODUCTION

For applications in complicated and high-stakes environments it can often mean operating in the minimal possible setting - that is with no access to knowledge of the environment dynamics nor intrinsic reward, nor even the ability to interact and test policies. In this case learning and inference must be done solely on the basis of logged trajectories from a competent demonstrator showing only the states visited and the the action taken in each case.

Clinical decision making is an important example of this, where there is great interest in learning policies from medical professionals but is completely impractical and unethical to deploy policies on patients mid-training. Moreover this is an area where it is not only the policies, but also knowledge of the demonstrator's preferences and goals, that we are interested in. While *imitation learning* (IL) generally deals with the problem of producing appropriate policies to match a demonstrator, with the added layer of understanding motivations this would then usually be approached through *inverse reinforcement learning* (IRL). Here attempting to learn the assumed underlying reward driving the demonstrator, before secondarily learning a policy that is optimal with respect to the reward using some forward *reinforcement learning* (RL) technique. By composing the RL and IRL procedures in order to perform IL we arrive at *apprenticeship learning* (AL), which introduces its own challenges, particularly in the offline setting. Notably for any given set of demonstrations there are (infinitely) many rewards for which the actions would be optimal (Ng et al., 2000). Max-margin (Abbeel & Ng, 2004) and max-entropy (Ziebart et al., 2008) methods for heuristically differentiating plausible rewards do so at the cost of potentially dismissing the *true* reward for not possessing desirable qualities. On the other hand a *Bayesian* approach to IRL (BIRL) is more conceptually satisfying, taking a probabilistic view of the reward, we are interested in the posterior distribution having seen the demonstrations (Ramachandran & Amir, 2007), accounting for all possibilities. BIRL is not without its own drawbacks though, as noted in Brown & Niekum (2019), making it inappropriate for modern complicated environments: assuming linear rewards; small, solvable environments; and repeated, inner-loop, calls to forward RL.

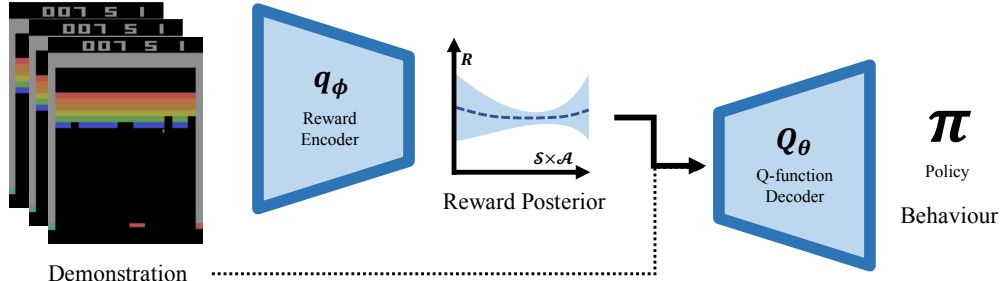

Figure 1: **Overview.** AVRIL is a framework for BIRL that works through an approximation in the variational Bayesian framework, considering the reward to be a latent representation of behaviour. A distribution over the reward, which is amortised over the demonstration space, is learnt that then informs an imitator $Q$-function policy. The dotted line represents a departure from a traditional auto-encoder as the input, alongside the latent reward, informs the decoder.

The main contribution then of this paper is a method for advancing BIRL beyond these obstacles, allowing for approximate reward inference using an arbitrarily flexible class of functions, in any environment, without costly inner-loop operations, and importantly entirely offline. This leads to our algorithm AVRIL, depicted in figure 1, which represents a framework for jointly learning a variational posterior distribution over the reward alongside an imitator policy in an *auto-encoder-esque* manner. In what follows we review the modern methods for offline IRL/IL (Section 2) with a focus on the approach of *Bayesian* IRL and the issues it faces when confronted with challenging environments. We then address the above issues by introducing our contributions (Section 3), and demonstrate the gains of our algorithm in real medical data and simulated control environments, notably that it is now possible to achieve Bayesian reward inference in such settings (Section 4). Finally we wrap up with some concluding thoughts and directions (Section 5). Code for AVRIL and our experiments is made available at `https://github.com/XanderJC/scalable-birl` and `https://github.com/vanderschaarlab/mlforhealthlabpub`.

## 2    APPROACHING APPRENTICESHIP AND IMITATION OFFLINE

**Preliminaries.**    We consider the standard Markov decision process (MDP) environment, with states $s \in \mathcal{S}$, actions $a \in \mathcal{A}$, transitions $T \in \Delta(\mathcal{S})^{\mathcal{S} \times \mathcal{A}}$, rewards $R \in \mathbb{R}^{\mathcal{S} \times \mathcal{A}1}$, and discount $\gamma \in [0, 1]$. For a policy $\pi \in \Delta(\mathcal{A})^{\mathcal{S}}$ let $\rho_{\pi}(s, a) = \mathbb{E}_{\pi, T}[\sum_{t=0}^{\infty} \gamma^t \mathbb{1}_{\{s_t=s, a_t=a\}}]$ be the induced unique occupancy measure alongside the state-only occupancy measure $\rho_{\pi}(s) = \sum_{a \in \mathcal{A}} \rho_{\pi}(s, a)$. Despite this full environment model, the only information available to us is the MDP$\backslash RT$, in that we have no access to either the underlying reward or the transitions, with our lacking knowledge of the transitions being also strong in the sense that further we are unable to simulate the environment to sample them. The learning signal is then given by access to $m$-many trajectories of some demonstrator assumed to be acting optimally w.r.t. the MDP, following a policy $\pi_D$, making up a data set $\mathcal{D}_{raw} = \{(s_1^{(i)}, a_1^{(i)}, \ldots, s_{\tau^{(i)}}^{(i)}, a_{\tau^{(i)}}^{(i)})\}_{i=1}^{m}$ where $s_t^{(i)}$ is the state and $a_t^{(i)}$ is the action taken at step $t$ during the $i$th demonstration, and $\tau^{(i)}$ is the (max) time horizon of the $i$th demonstration. Given the Markov assumption though it is sufficient and convenient to consider the demonstrations simply as a collection of $n$-many state, action, next state, next action tuples such that $\mathcal{D} = \{(s_i, a_i, s_i', a_i')\}_{i=1}^{n}$ with $n = \sum_{i=1}^{m}(\tau^{(i)} - 1)$.

**Apprenticeship through rewards.**    Typically AL proceeds by first inferring an appropriate reward function with an IRL procedure (Ng et al., 2000; Ramachandran & Amir, 2007; Rothkopf & Dimitrakakis, 2011; Ziebart et al., 2008) before running forward RL to obtain an appropriate policy. This allows for easy mix-and-match procedures, swapping in different standard RL and IRL methods depending on the situation. These algorithms though depend on either knowledge of $T$ in order to solve exactly or the ability to perform roll-outs in the environment, with little previous work focusing on the entirely offline setting. One simple solution is through attempting to learn the dynamics

---

[1]We define a state-action reward here, as is usual in the literature. Extensions to a state-only reward are simple, and indeed can be preferable, as we will see later.

(Herman et al., 2016), though without a large supply of diverse demonstrations or a small environment this becomes impractical given imperfections in the model. Alternatively Klein et al. (2011) and Lee et al. (2019) attempt off-policy feature matching through least-squared temporal difference and deep neural networks to uncover appropriate feature representations.

**Implicit-reward policy learning.** Recent work has often forgone an explicit representation of the reward. Moving within the maximum-entropy RL framework (Ziebart, 2010; Levine, 2018), Ho & Ermon (2016) noted that the full procedure (RL ∘ IRL) can be interpreted equivalently as the minimisation of some divergence between occupancy measures of the imitator and demonstrator:

$$\arg\min_{\pi}\{\psi^*(\rho_\pi - \rho_{\pi_D}) - H(\pi)\}, \tag{1}$$

with $H(\pi)$ being the discounted causal entropy (Bloem & Bambos, 2014) of the policy and $\psi^*$ the Fenchel conjugate of a chosen regulariser on the form of the reward. These are typically optimised in an adversarial fashion (Goodfellow et al., 2014) and given the focus on evaluating $\rho_\pi$ this often requires extensive interaction with the environment, otherwise banking on approximations over a replay buffer (Kostrikov et al., 2018) or a reformulation of the divergence to allow for off-policy evaluation (Kostrikov et al., 2019). Bear in mind that optimal policies within the maximum-entropy framework are parameterised by a Boltzmann distribution:

$$\pi(a|s) = \frac{\exp(Q(s,a))}{\sum_{b\in\mathcal{A}}\exp(Q(s,b))}, \tag{2}$$

with $Q(s,a)$ the *soft* $Q$-function, defined recursively via the *soft* Bellman-equation:

$$Q(s,a) \triangleq R(s,a) + \gamma\mathbb{E}_{s'\sim\rho_\pi}\left[\operatorname*{soft\,max}_{a'} Q(s',a'))\right]. \tag{3}$$

Then for a learnt parameterised policy given in terms of $Q$-values from a function approximator $Q_\theta$, we can obtain an *implied* reward given by:

$$R_{Q_\theta}(s,a) = Q_\theta(s,a) - \gamma\mathbb{E}_{s'\sim\rho_\pi}\left[\log\left(\sum_{a'\in\mathcal{A}}\exp(Q_\theta(s',a'))\right)\right]. \tag{4}$$

A number of algorithms make use of this fact with Piot et al. (2014) and Reddy et al. (2019) working by essentially placing a sparsity prior on this implied reward, encouraging it towards zero, and thus incorporating subsequent state information. Alternatively Jarrett et al. (2020) show that even the simple behavioural cloning (Bain & Sammut, 1995) is implicitly maximising some reward with an approximation that the expectation over states is taken with respect to the demonstrator, not the learnt policy. They then attempt to rectify part of this approximation using the properties of the energy-based model implied by the policy (Grathwohl et al., 2019).

The problem with learning an implicit reward in an offline setting is that it remains just that, *implicit*, only able to be evaluated at points seen in the demonstrations, and even then only approximately. Thus even if their consideration improves imitator policies performance they offer no real improvement for interpretation.

## 2.1 Bayesian Inverse Reinforcement Learning

We are then resigned to directly reason about the underlying reward, bringing us back to the question of IRL, and in particular BIRL for a principled approach to reasoning under uncertainty. Given a prior over possible functions, having seen some demonstrations, we calculate the posterior over the function using a theoretically simple application of Bayes rule. Ramachandran & Amir (2007) defines the likelihood of an action at a state as a Boltzmann distribution with inverse temperature and respective state-action values, yielding a probabilistic demonstrator policy given by:

$$\pi_D(a|s, R) = \frac{\exp(\beta Q_R^{\pi_D}(s,a))}{\sum_{b\in\mathcal{A}}\exp(\beta Q_R^{\pi_D}(s,b))}, \tag{5}$$

where $\beta \in [0, \infty)$ represents the the confidence in the optimality of the demonstrator. Note that despite similarities, moving forward we are no longer within the maximum-entropy framework and $Q_R^\pi(s,a)$ now denotes the traditional, not soft (as in equation 3), state-action value ($Q$-value) function given a reward $R$ and policy $\pi$ such that $Q_R^\pi(s,a) = \mathbb{E}_{\pi,T}[\sum_{t=0}^{\infty}\gamma^t R(s_t)|s_0 = s, a_0 = a]$. Unsurprisingly this yields an intractable posterior distribution leading to a Markov chain Monte Carlo (MCMC) algorithm based on a random grid-walk to sample from the posterior.

**Issues in complex and unknown environments.** This original formulation, alongside extensions that consider maximum-a-posteriori inference (Choi & Kim, 2011) and multiple rewards (Choi & Kim, 2012; Dimitrakakis & Rothkopf, 2011), suffer from three major drawbacks that make them impractical for modern, complicated, and model-free task environments.

1. *The reward is a linear combination of state features.* Naturally this is a very restrictive class of functions and assumes access to carefully hand-crafted features of the state space.

2. *The cardinality of the state-space is finite, $|\mathcal{S}| < \infty$.* Admittedly this can be relaxed in practical terms, although it does mean the rapid-mixing bounds derived by Ramachandran & Amir (2007) do not hold at all in the infinite case. For finite approximations they scale at $\mathcal{O}(|\mathcal{S}|^2)$, rapidly becoming vacuous and causing BIRL to inherit the usual MCMC difficulties on assessing convergence and sequential computation (Gamerman & Lopes, 2006).

3. *The requirement of an inner-loop MDP solve.* Most importantly at every step a new reward is sampled and the likelihood of the data must then be evaluated. This requires calculating the $Q$-values of the policy with respect to the reward, in other words running forward RL. While not an insurmountable problem in the simple cases where everything is known and can be quickly solved with a procedure guaranteed to converge correctly, this becomes an issue in the realm where only deep function approximation works adequately (i.e. the non-tabular setting). DQN training for example easily stretches into hours (Mnih et al., 2013) and will have to be repeated thousands of times, making it completely untenable.

We have seen that even in the most simple setting the problem of exact Bayesian inference over the reward is intractable, and the above limitations of the current MCMC methods are not trivial to overcome. Consequently very little work has been done in the area and there still remain very open challenges. Levine et al. (2011) addressed linearity through a Gaussian process approach, allowing for a significantly more flexible and non-linear representation though introducing issues of its own, namely the computational complexity of inverting large matrices (Rasmussen, 2003). More recently Brown & Niekum (2019) have presented the only current solution to the inner-loop problem by introducing an alternative formulation of the likelihood, one based on human recorded pairwise preferences over demonstrations that significantly reduces the complexity of likelihood. However labelled preferences certainly can't be assumed always available and while very effective for the given task is not appropriate in the general case. One of the key aspects of our contribution is that we are able to deal with all three of these issues while also not requiring any additional information.

**The usefulness of uncertainty.** On top of the philosophical consistency of Bayesian inference there are a number of reasons for wanting a measure of uncertainty over any uncovered reward that are not available from regular IRL algorithms. First that the (epistemic) uncertainty revealed by Bayesian inference tells us a lot about what areas of the state-space we really cannot say anything about because we haven't seen any demonstrations there - potentially informing future data collection if that is possible (Mindermann et al., 2018). Additionally in the cases we are mostly concerned about (e.g. medicine) we have to be very careful about letting algorithms pick actions in practice and we are interested in performing *safe* or *risk-averse* imitation, for which a degree of confidence over learnt rewards is necessary. Brown et al. (2020) for example use a distribution over reward to optimise a conditional value-at-risk instead of expected return so as to bound potential downsides.

## 3  APPROXIMATE VARIATIONAL REWARD IMITATION LEARNING

**A variational Bayesian approach.** In this section we detail our method, AVRIL, for efficiently learning an imitator policy and performing reward inference simultaneously. Unlike the previously mentioned sampling or MAP-based methods, we employ variational inference (Blei et al., 2017) to reason about the posterior. Here we posit a surrogate distribution $q_\phi(R)$, parameterised by $\phi$, and aim to minimise the Kullback-Leibler (KL) divergence to the posterior, resulting in an objective:

$$\min_\phi \{ D_{\mathrm{KL}}(q_\phi(R)||p(R|\mathcal{D})) \}. \tag{6}$$

This divergence is still as troubling as the posterior to evaluate, leading to an auxiliary objective function in the Evidence Lower BOund (ELBO):

$$\mathcal{F}(\phi) = \mathbb{E}_{q_\phi}\big[\log p(\mathcal{D}|R)\big] - D_{KL}\big(q_\phi(R)||p(R)\big), \tag{7}$$

where it can be seen that maximisation over $\phi$ is equivalent to (6). We are agnostic towards the form of both the prior and variational distribution, for simplicity here we assume a Gaussian process prior with mean zero and unit variance over $R$ alongside the variational posterior distribution given by:

$$q_\phi(R) = \mathcal{N}(R; \mu, \sigma^2), \tag{8}$$

where $\mu, \sigma^2$ are the outputs of an encoder neural network taking $s$ as input and parameterised by $\phi$. Note that for the algorithm that we will describe these choices are not a necessity and can be easily substituted for more expressive distributions if appropriate. Maintaining the assumption of Boltzmann rationality on the part of the demonstrator, our objective takes the form:

$$\mathcal{F}(\phi) = \mathbb{E}_{q_\phi}\left[\sum_{(s,a)\in\mathcal{D}} \log \frac{\exp(\beta Q_R^{\pi_D}(s,a))}{\sum_{b\in\mathcal{A}} \exp(\beta Q_R^{\pi_D}(s,b))}\right] - D_{KL}\big(q_\phi(R)||p(R)\big). \tag{9}$$

The most interesting (and problematic) part of this objective as ever centres on the evaluation of $Q_R^{\pi_D}(s,a)$. Notice that what is really required here is an expression of the $Q$-values as a smooth function of the reward such that with samples of $R$ we could take gradients w.r.t. $\phi$. Of course there is little hope of obtaining this simply, by itself it is a harder problem than that of forward RL which only attempts to evaluate the $Q$-values for a specific $R$ and already in complicated environments has to rely on function approximation and limited guarantees.

A naive approach would be to sample $\hat{R}$ and then approximate the $Q$-values with a second neural network, solving offline over the batched data using a least-squared TD/$Q$-learning algorithm, as is the approach forced on sampling based BIRL methods. It is in fact though doubly inappropriate for this setting, not only does this require a solve as an inner-loop but importantly differentiating through the solving operation is extremely impractical, it requires backpropagating through a number of gradient updates that are essentially unbounded as the complexity of the environment increases.

**A further approximation.** This raises an important question - is it possible to jointly optimise a policy and variational distribution only once instead of requiring a repeated solve? This is theoretically suspect, the $Q$-values are defined on a singular reward, constrained as $R(s,a) = \mathbb{E}_{s',a'\sim\pi,T}[Q_R^\pi(s,a) - \gamma Q_R^\pi(s',a')]$ so we cannot learn a particular standard $Q$-function that reflects the entire distribution. But can we learn a policy that reflects the expected reward using a second policy neural network $Q_\theta$? We can't simply optimise $\theta$ alongside $\phi$ to maximise the ELBO though as that completely ignores the fact that the learnt policy is intimately related to the distribution over the reward. Our solution to ensure then that they behave as intended is by constraining $q_\phi$ and $Q_\theta$ to be consistent with each other, specifically that the implied reward of the policy is sufficiently likely under the variational posterior (equivalently that the negative log-likelihood is sufficiently low). Thus we arrive at a constrained optimisation objective given by:

$$\max_{\phi,\theta} \sum_{(s,a)\in\mathcal{D}} \log \frac{\exp(\beta Q_\theta(s,a))}{\sum_{b\in\mathcal{A}} \exp(\beta Q_\theta(s,b))} - D_{KL}\big(q_\phi(R)||p(R)\big), \tag{10}$$

$$\text{subject to } \mathbb{E}_{\pi,T}[-\log q_\phi(Q_\theta(s,a) - \gamma Q_\theta(s',a'))] < \epsilon.$$

with $\epsilon$ reflecting the strength of the constraint. Rewriting (10) as a Lagrangian under the KKT conditions (Karush, 1939; Kuhn & Tucker, 1951), and given complimentary slackness, we obtain a practical objective function:

$$\mathcal{F}(\phi,\theta,\mathcal{D}) = \sum_{(s,a,s',a')\in\mathcal{D}} \log \frac{\exp \beta Q_\theta(s,a)}{\sum_{b\in\mathcal{A}} \exp(\beta Q_\theta(s,b))} - D_{KL}\big(q_\phi(R(s,a))||p(R(s,a))\big)$$
$$+ \lambda \log q_\phi(Q_\theta(s,a) - \gamma Q_\theta(s',a')). \tag{11}$$

Here the KL divergence between processes is approximated over a countable set, and $\lambda$ is introduced to control the strength of constraint.

**On the implementation.** Optimisation is simple as both networks are maximising the same objective and gradients can be easily obtained through backpropagation while being amenable to mini-batching, allowing you to call your favourite gradient-based stochastic optimisation scheme. We re-iterate though that AVRIL really represents a framework for doing BIRL and not a specific model

---

**Algorithm 1:** Approximate Variational Reward Imitation Learning (AVRIL)

---

**Result:** Parameters $\phi$ of variational distribution and $\theta$ of policy Q-function
**Input:** $\mathcal{D}, S, A, \gamma, \lambda$, learning rate $\eta$, mini-batch size $b$;
Initialise $\phi, \theta$ ;                 ▷ Can concatenate into single vector $(\phi, \theta)$
**while** *not converged* **do**
    Sample $\mathcal{D}_{mini}$ from $\mathcal{D}$ ;
    $\mathcal{F}(\phi, \theta, \mathcal{D}) = \mathbb{E}[\frac{n}{b}\mathcal{F}(\phi, \theta, \mathcal{D}_{mini})]$ ;                 ▷ MC estimate total loss
    $(\phi', \theta') \leftarrow (\phi, \theta) + \eta\nabla_{\phi,\theta}\mathcal{F}(\phi, \theta, \mathcal{D})$ ;                 ▷ Gradient step for $\phi, \theta$
    $\phi, \theta \leftarrow \phi', \theta'$
**end**
**Return:** $\phi, \theta$

---

since $Q_\theta$ and $q_\phi$ represent arbitrary function approximators. So far we have presented both as neural networks, but this does not have to be the case. Of course the advantage of them is their flexibility and ease of training but they are still inherently black box. It is then perfectly possible to swap in any particular function approximator if the task requires it, using simple linear models for example may slightly hurt performance but allow for more insight. Despite the specific focus on infinite state-spaces, AVRIL can still even be applied in the tabular setting by simply representing the policy and variational distribution with multi-dimensional tensors. Having settled on their forms, equation (11) is calculated simply and the joint gradient with respect to $\theta$ and $\phi$ is straight-forwardly returned using any standard auto-diff package. The whole process is summarised in Algorithm 1.

We can now see how AVRIL does not suffer the issues outlined in section 2.1. Our form of $q_\phi(R)$ is flexible and easily accommodates a non-linear form of the reward given a neural architecture - this also removes any restriction on $\mathcal{S}$, or at least allows for any state space that is commonly tackled within the IL/RL literature. Additionally we have a single objective for which all parameters are maximised simultaneously - there are no inner-loops, costly or otherwise, meaning training is faster than the MCMC methods by a factor equal roughly to the number of samples they would require.

**The generative model view.** Ultimately a policy represents a generative model for the behavioural data we see. Ho & Ermon (2016) explicitly make use of this fact by casting the problem in the GAN framework (Goodfellow et al., 2014). Our method is more analogous to a VAE (Kingma & Welling, 2013), though not exactly, where given the graphical model in figure 2 the reward can be seen as a latent representation of the policy. Our approach takes the seen data and amortises the inference, *encoding* over the state space. The policy does not act as a *decoder* in precisely taking any given encoded reward and outputting a policy, but it does take the whole reward posterior and translate it into actions and therefore

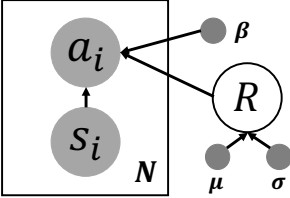

Figure 2: Graphical model for Bayesian IRL

behaviour. This approach has its advantages, in both meaningful interpretation of the latent reward (which is non-existent in adversarial methods), and that we forgo the practical difficulties of alternating *min-max* optimisation (Kodali et al., 2017) while maintaining a generative view of the policy.

**Temporal consistency through reward regularisation.** Considering only the first term of (11) yields the standard behavioural cloning setup (where the logits output can be interpreted as the $Q$-values) as it removes the reward from the equation and just focuses on matching actions to states. AVRIL can then be seen as a policy-learning method regularised by the need for the implied reward to be consistent. Note that this does not induce any necessary bias since the logits normally contain an extra degree of freedom allowing them to arbitrarily shift by some scale factor. This factor is now explicitly constrained by giving the logits additional meaning in that they represent $Q$-values. This places great importance on the KL term, since every parameterisation of a policy will have an associated implied reward, the KL regularises these to be not so far from the prior and preventing the reward from overfitting to the policy and becoming pointless. It also is able to double as a regularising term in a similar manor to previous reward-regularisation methods (Piot et al., 2014; Reddy et al., 2019) depending on the chosen prior, encouraging the reward to be close to zero:

**Proposition 1 (Reward Regularisation)** Assume that the constraint in (10) is satisfied in that $\mathbb{E}_{q_\phi}[R(s,a)] = \mathbb{E}_{\pi,T}[Q_\theta(s,a) - \gamma Q_\theta(s',a')]$, then given a standard normal prior $p(R) = \mathcal{N}(R;0,1)$ the KL divergence yields a sparsity regulator on the implied reward:

$$\mathcal{L}_{reg} = \sum_{(s,a,s',a') \in \mathcal{D}} \frac{1}{2}\big(Q_\theta(s,a) - \gamma Q_\theta(s',a')\big)^2 + g(\text{Var}_{q_\phi}[R(s,a)]). \tag{12}$$

*Proof.* Appendix. □ This follows immediately from the fact that the divergence evaluates as $D_{KL}\big(q_\phi(R(s,a))||p(R(s,a))\big) = \frac{1}{2}(-\log(\text{Var}_{q_\phi}[R(s,a)]) - 1 + \text{Var}_{q_\phi}[R(s,a)] + \mathbb{E}_{q_\phi}[R(s,a)]^2)$

This then allows AVRIL to inherit the benefit of these methods while also explicitly learning a reward that can be queried at any point. We are also allowed the choice of whether it is state-only or state-action. This has so far been arbitrary, but it is important to consider that a state-only reward is a necessary and sufficient condition for a reward that is fully disentangled from the dynamics (Fu et al., 2018). Thus by learning such a reward and given the final term of (11) that directly connects one-step rewards in terms of the policy, this forces the policy (not the reward) to account for the dynamics of the system ensuring temporal consistency in a way that BC for example simply can't. Alternatively using a state-action reward means that inevitably some of the temporal information leaks out of the policy and into the reward - ultimately to the detriment of the policy but potentially allowing for a more interpretable (or useful) form of reward depending on the task at hand.

## 4 EXPERIMENTS

**Experimental setup.** We are primarily concerned with the case of medical environments, which is exactly where the issue of learning without interaction is most crucial, you just cannot let a policy sample treatments for a patient to try to learn more about the dynamics. It is also where a level of interpretability in what has been learnt is important, since the consequence of actions are potentially very impactful on human lives. As such we focus our evaluation on learning on a real-life healthcare problem, with demonstrations taken from the Medical Information Mart for Intensive Care (MIMIC-III) dataset (Johnson et al., 2016). The data contains trajectories of patients in intensive care recording their condition and theraputic interventions at one day intervals. We evaluate the ability of the methods to learn a medical policy in both the two and four action setting - specifically whether the patient should be placed on a ventilator, and the decision for ventilation in combination with antibiotic treatment. These represent the two most common, and important, clinical interventions recorded in the data. Without a recorded notion of reward, performance is measured with respect to *action matching* against a held out test set of demonstrations with cross-validation.

Alongside the healthcare data and for the purposes of demonstrating generalisability, we provide additional results on standard environments of varying complexity in the RL literature, the standard control problems of: CartPole, a classic control environment aiming to swing up and balance a pendulum; Acrobot, which aims to maintain a sequence of joints above a given height; and LunarLander, guiding a landing module to a safe touchdown on the moon surface. In these settings given sufficient demonstration data all benchmarks are very much capable of reaching demonstrator level performance, so we test the algorithms on their ability to handle sample complexity in the low data regime by testing their performance when given access to a select number of trajectories which we adjust, replicating the setup in Jarrett et al. (2020). With access to a simulation through the OpenAI gym (Brockman et al., 2016), we measure performance by deploying the learnt policies live and calculating their average return over 300 episodes.

**Benchmarks.** We test our method (**AVRIL**) against a number of benchmarks from the offline IRL/IL setting: Deep Successor Feature Network (**DSFN**) (Lee et al., 2019), an offline adaptation of max-margin IRL that generalises past the linear methods using a deep network with least-squares temporal-difference learning, the only other method that produces both a reward and policy; Reward-regularized Classification for Apprenticeship Learning (**RCAL**) (Piot et al., 2014), where an explicit regulariser on the sparsity of the implied reward is introduced in order to account for the dynamics information; ValueDICE (**VDICE**) (Kostrikov et al., 2019), an adversarial imitation learning, adapted for the offline setting by removing the replay regularisation; Energy-based Distribution Matching (**EDM**) (Jarrett et al., 2020), the state-of-the-art in offline imitation learning; and finally the standard example of Behavioural Cloning (**BC**). To provide evidence that we are indeed

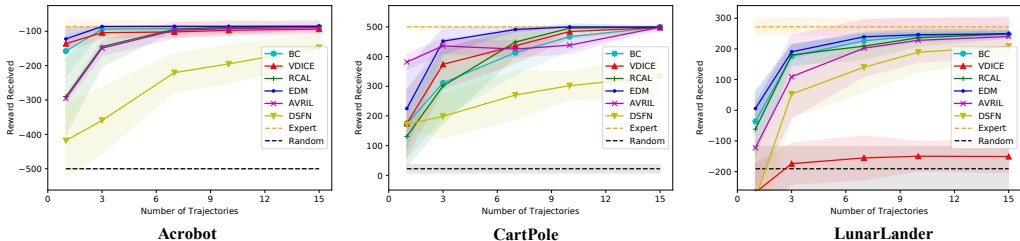

Figure 3: **Control environments performance**. We plot the average returns received by the policies when deployed live in the environment against the number of trajectories seen during training.

Table 1: **Healthcare performance.** Comparison of methods on the MIMIC-III dataset. Performance of the policy is evaluated on the quality of *action matching* against a held out test set of demonstrations. We report the accuracy (ACC), area under the receiving operator characteristic curve (AUC) and average precision score (APS).

| Metric | Ventilator | | | Ventilator + Antibiotics | | |
|---|---|---|---|---|---|---|
| | ACC | AUC | APS | ACC | AUC | APS |
| BC | $0.873 \pm 0.007$ | $0.916 \pm 0.002$ | $0.904 \pm 0.003$ | $0.700 \pm 0.009$ | $0.864 \pm 0.003$ | $0.665 \pm 0.009$ |
| VDICE | $0.879 \pm 0.002$ | $0.915 \pm 0.002$ | $0.904 \pm 0.003$ | $0.710 \pm 0.005$ | $0.863 \pm 0.002$ | $0.675 \pm 0.004$ |
| RCAL | $0.870 \pm 0.012$ | $0.916 \pm 0.003$ | $0.904 \pm 0.005$ | $0.702 \pm 0.008$ | $0.865 \pm 0.004$ | $0.669 \pm 0.006$ |
| DSFN | $0.869 \pm 0.005$ | $0.905 \pm 0.003$ | $0.885 \pm 0.001$ | $0.683 \pm 0.007$ | $0.856 \pm 0.002$ | $0.670 \pm 0.004$ |
| EDM | $0.882 \pm 0.011$ | $\mathbf{0.920 \pm 0.002}$ | $0.909 \pm 0.003$ | $0.716 \pm 0.008$ | $0.873 \pm 0.002$ | $0.682 \pm 0.004$ |
| A-RL | $0.875 \pm 0.010$ | $0.904 \pm 0.002$ | $0.927 \pm 0.002$ | $0.718 \pm 0.010$ | $0.864 \pm 0.002$ | $0.665 \pm 0.005$ |
| AVRIL | $\mathbf{0.891 \pm 0.002}$ | $0.917 \pm 0.001$ | $\mathbf{0.940 \pm 0.001}$ | $\mathbf{0.754 \pm 0.001}$ | $\mathbf{0.884 \pm 0.000}$ | $\mathbf{0.708 \pm 0.002}$ |

learning an appropriate reward we show an ablation of our method on the MIMIC data: we take the reward learnt by AVRIL and use it as the 'true' reward used to train a $Q$-network offline to learn a policy (**A-RL**). Note that we have not included previous BIRL methods for the reasons explained in section 2.1, training a network just once in these environments takes in the order of minutes and repeating this sequentially thousands of times is just not practical. For aid in comparison all methods share the same network architecture of two hidden layers of 64 units with ELU activation functions and are trained using Adam (Kingma & Ba, 2014) with learning rates individually tuned. Further details on experimental setup and the implementation of benchmarks can be found in the appendix.

**Evaluation.** We see for all tasks AVRIL learns an appropriate policy that performs strongly across the board, being competitive in all cases and in places beating out all of the other benchmarks. The results for our healthcare example are given in table 1, with AVRIL performing very strongly, having the highest accuracy and precision score in both tasks. The results for the control environments are shown in figure 3. AVRIL performs competitively and is easily capable of reaching demonstrator level performance in the samples given for these tasks, though not always as quickly as some of the dedicated offline IL methods.

**Reward insight.** Remember though that task performance is not exactly our goal. Rather the key aspect of AVRIL is the inference over the unseen reward in order to gain information about the preferences of the agent that other black-box policy methods can't. In the previous experiments our reward encoder was a neural network for maximum flexibility and we can see from the performance of A-RL we learn a representation of the reward that can be used to relearn in the environment very effectively, albeit not quite to the same standard of AVRIL. Note this also reflects an original motivation for AVRIL in that offpolicy RL on top of a learnt reward suffers. In figure 4 we explore how to gain more insight from the learnt reward using different parameterisations of the reward. The top graph shows how a learnt state-action reward changes as a function of blood-oxygen level for an otherwise healthy patient, and it can be seen that as it drops below average the reward for ventilating the patient becomes much higher (note this is average for patients *in* the ICU, not across the general population). While this is intuitive we still have to query a neural network repeatedly over the state space to gain insight, the bottom graph of figure 4 presents then a simpler but perhaps more useful

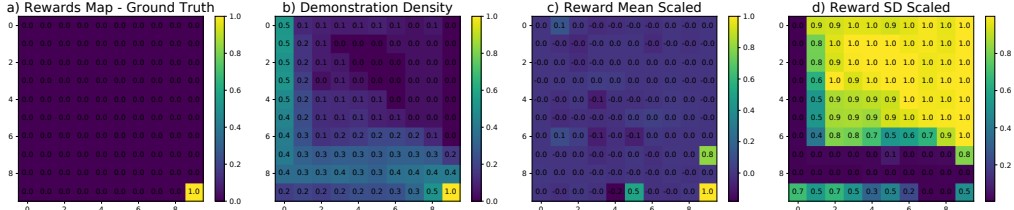

Figure 5: **Gridworld example**. Scaled heat-maps of: the ground truth reward; the relative state occupancy of the expert demonstrations; the reward posterior mean; and reward standard deviation.

representation. In this case we learn a state-only reward as before but as a linear model. This is not as strong a constraint on the policy since that is still free to be non-linear as a neural network but simultaneously allows us the insight of what our model considers high value in the environment as we plot the relative model coefficients for each covariate. We can see here for example that the biggest impact on the overall estimated quality of a state is given by blood pressure, well known as an important indicator of health (Hepworth et al., 1994), strongly impacted by trauma and infection.

**Gridworld ground-truth comparison** While environments like MIMIC are the main focus of this work they do not lend them selves to inspection of the uncovered reward as the ground truth simply is not available to us. We thus demonstrate on a toy gridworld environment, in order to clearly see the effect of learning a posterior distribution over the reward. In this (finite) example both the encoder and decoder are represented by tensors but otherwise the procedure remains the same. Figure 5 plots scaled heat-maps of: a) the ground truth reward; b) the relative state occupancy of the expert demonstrations, obtained using value-iteration; c) the reward posterior mean; and d) the reward standard deviation. The interesting thing to note is that the standard deviation of the learnt reward essentially resembles the complement of the state occupancy - revealing the epistemic uncertainty around that part of the state-space given we haven't seen any demonstrations there.

## 5 CONCLUSIONS

We have presented a novel algorithm, Approximate Variational Reward Imitation Learning, for addressing the scalability issues that prevent current Bayesian IRL methods being used in large and unknown environments. We show that this performs strongly on real and toy data for learning imitation policies completely offline and importantly recovers a reward that is both effective for retraining policies but also offers useful insight into the preferences of the demonstrator. Of course this still represents an approximation, and there is room for further, more exact methods or else guarantees on the maximum divergence. We have focused on simply obtaining the appropriate uncertainty over reward as well as imitation in high stakes environments - in these settings it is crucial that learnt policies avoid catastrophic failure and so how exactly to use the uncertainty in order to achieve truly *safe* imitation (or indeed better-that-demonstrator apprenticeship) is increasingly of interest.

Figure 4: **(Top)** A state-action reward is learnt and plotted for an otherwise average patient as their blood oxygen level changes. **(Bottom)** The associated weights given a state-only reward as a linear function of the state-space.

## ACKNOWLEDGEMENTS

AJC would like to acknowledge and thank Microsoft Research for its support through its PhD Scholarship Program with the EPSRC. This work was additionally supported by the Office of Naval Research (ONR) and the NSF (Grant number: 1722516). We would like to thank all of the anonymous reviewers on OpenReview, alongside the many members of the van der Schaar lab, for their input, comments, and suggestions at various stages that have ultimately improved the manuscript.

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

## A  EXPERIMENTAL SETUP

**Expert Demonstrators.**  Demonstrations are produced by running pre-trained and hyperparmeter-optimised agents taken from the RL Baselines Zoo (Raffin, 2018) in OpenAI Stable Baselines (Hill et al., 2018). For Acrobot and LunarLander these are DQNs (Mnih et al., 2013), while CartPole uses PPO2 (Schulman et al., 2017). Trajectories were then sub-sampled for every 20th step in Acrobot and CartPole, and every 5th step in LunarLander.

**Testing setup.**  For control environments algorithms were presented with (1,3,7,10,15) trajectories uniformly sampled from a pool of 1000 expert trajectories. Each algorithm was then trained until convergence and tested by performing 300 live roll-outs in the simulated environment and recording the average accumulated reward received in each episode. This whole process was then repeated 10 times, consequently with different initialisations and seen trajectories.

**Implementations.**  All methods are neural network based and so in experiments they share the same architecture of 2 hidden layers of 64 units each connected by exponential linear unit (ELU) activation functions.

Publicly available code was used in the implementations of a number of the benchmarks, specifically:

- VDICE (Kostrikov et al., 2019):
  ```
  https://github.com/google-research/google-research/tree/
  master/value_dice
  ```

- DSFN (Lee et al., 2019):
  ```
  https://github.com/dtak/batch-apprenticeship-learning
  ```

- EDM (Jarrett et al., 2020):
  ```
  https://github.com/wgrathwohl/JEM
  ```

Note that VDICE was originally designed for continuous actions with a Normal distribution output which we adapt for the experiments by replacing with a Gumbel-softmax.

## B  PROOFS

**Proof of proposition 1.**  Assuming the constraint is satisfied, we are maximising the following objective:

$$\mathcal{F}(\phi, \theta) = \sum_{(s,a,s',a') \in \mathcal{D}} \log \frac{\exp \beta Q_\theta(s,a))}{\sum_{b \in \mathcal{A}} \exp(\beta Q_\theta(s,b))} - D_{KL}\big(q_\phi(R(s,a))||p(R(s,a))\big) \qquad (13)$$

Which is equivalent to minimising the negative value

$$\mathcal{F}(\phi, \theta) = \sum_{(s,a,s',a') \in \mathcal{D}} \underbrace{-\log \frac{\exp \beta Q_\theta(s,a))}{\sum_{b \in \mathcal{A}} \exp(\beta Q_\theta(s,b))}}_{\mathcal{L}_{BC}} + \underbrace{D_{KL}\big(q_\phi(R(s,a))||p(R(s,a))\big)}_{\mathcal{L}_{reg}}, \qquad (14)$$

with the first term $\mathcal{L}_{BC}$ being the negative log-likelihood of the data and the classic behavioural cloning objective. Now given a standard Gaussian prior then the KL divergence of a Gaussian with mean $\mu$ and variance $\sigma^2$ from the prior is given by $\frac{1}{2}(-\log(\sigma^2) + \sigma^2 - 1 + \mu^2)$ (Kingma & Welling,

2013). Then given our prior $p(R(s,a)) = \mathcal{N}(R; 0, 1)$, the KL evaluates as:

$$\mathcal{L}_{reg} = \sum_{(s,a,s',a')\in\mathcal{D}} D_{KL}\big(q_\phi(R(s,a))||p(R(s,a))\big) \tag{15}$$

$$= \sum_{(s,a,s',a')\in\mathcal{D}} \frac{1}{2}(-\log(\text{Var}_{q_\phi}[R(s,a)]) + \text{Var}_{q_\phi}[R(s,a)] - 1 + \mathbb{E}_{q_\phi}[R(s,a)]^2) \tag{16}$$

$$= \sum_{(s,a,s',a')\in\mathcal{D}} \frac{1}{2}(\mathbb{E}_{q_\phi}[R(s,a)]^2) + g(\text{Var}_{q_\phi}[R(s,a)]) \tag{17}$$

$$= \sum_{(s,a,s',a')\in\mathcal{D}} \frac{1}{2}\big(Q_\theta(s,a) - \gamma Q_\theta(s',a')\big)^2 + g(\text{Var}_{q_\phi}[R(s,a)]) \tag{18}$$

Since by assumption $\mathbb{E}_{q_\phi}[R(s,a)] = \mathbb{E}_{\pi,T}[Q_\theta(s,a) - \gamma Q_\theta(s',a')]$ with the expectation approximated over samples in the data and considering a definition of the function $g$ to be $g(\text{Var}_{q_\phi}[R(s,a)]) = \frac{1}{2}(-\log(\text{Var}_{q_\phi}[R(s,a)]) + \text{Var}_{q_\phi}[R(s,a)] - 1)$. $\square$

