# OpenReview forum: "Scalable Bayesian Inverse Reinforcement Learning"
_ICLR.cc/2021/Conference — ICLR 2021 Poster_

### Official Review · AnonReviewer4 · 2020-10-24
**The authors propose two neural networks to solve the inverse RL tasks. These are expected to resemble an autoencoder. One of the neural networks learns a policy that can recover the decisions made in the demonstrations, and the second neural network learns the reward of the policy that is learned by the first neural network. The empirical results are promissing with respect to the "action matching". The reward learned by the algorithm is not compared with any other approaches.**

**Rating:** 7
**Confidence:** 4

**Review:**

Sequential decision-making is investigated in this paper. A number of demonstrations (i.e. trajectories) is provided to the algorithm that has to lean a policy using only data. Any exploration using sampling from the environment is not possible. To allow for a better understanding of the domain, the algorithm also has to recover the reward that is optimized by the agent that provides demonstrations. In this paper, the authors propose two neural networks that are expected to resemble an autoencoder. One of the neural networks learns a policy that can recover the decisions made in the demonstrations, and the second neural network learns the reward of the policy that is learned by the first neural network. The empirical results show that the new method outperforms alternative methods with respect to the "action matching". The reward learned by the algorithm is not compared with any other approaches.

The overall quality of writing in this paper is at a high, professional level. The authors know related literature, and citations are relevant and sufficient. I noticed very few types that are below.

The approach proposed in the paper is very interesting, and this work could be significant because the medical application requires this kind of method. I am not in position to give a strong recommendation to accept this paper because I feel that the core algorithm is not explained sufficiently in the current version of the paper, and a few important experiments are missing. This position is justified by my major concerns below.

Major concerns:

The biggest problem that I have is that I cannot easily link Fig. 1 with Eq. (11) because the current explanations in the paper leave many details implied. In particular, Fig. 1, shows two neural networks, one for $q_\phi$ and one for $Q_\theta$, but the paper does not explain what are the inputs and what are the outputs in both networks, and which components of Eq. (11) are used to optimize every network. It seems that the learning objective in $Q_\theta$ uses terms 1 and 3 in Eq. (11); this is possible when $\phi$ are frozen at the time when $\theta$ are updated. This must be a coordinate ascent algorithm then. So, the reward predicted by $q_\phi$ is not the input to $Q_\theta$; it is used only as a penalty term in $Q_\theta$. This is confusing because Fig. 1 indicates that R is the input.

Next, let's consider $q_\phi$. It takes (s,a) tuples as input, and it predicts $(\mu,\sigma)$ that define the normal distribution over R for every (s,a). I guess that terms 2 and 3 in Eq. (11) are used as objectives. The KL divergence wants the reward to be close to the prior, and the 3rd term, wants the predictions made by $q_\phi$ to be close to the real differences encoded by $Q_\theta$. So, again, when $\theta$s are fixed, the $\phi$ can be updated. The parameters of one of the networks can be updated, when the parameters of the other network are frozen.

Thus, this summary of the two networks indicates that this procedure is not an autoencoder. There are two independent NNs here, that are trained using coordinate ascent. In particular, if $Q_\theta$ is the decoder, then its error should be backpropagated to the second network, i.e., to $q_\phi$, or at least the decoder should be decoding using the low-dimensional embeddings instead of receiving entire input. This is definitely not the case here. There are two networks that are trained independently, i.e. they don't share the error information during their own backpropagations. I think that these nuances of the algorithms should be made clear; it took me some time to identify these details.

I would further claim that $q_\phi$ is not even required to learn $Q_\theta$ with good predictive accuracy. It would be sufficient to use $Q_\theta$ with the first term in Eq. (11) and then penalize the differences in Q(s,a) that are in the second term without the use of $q_\phi$ at all. $Q_\theta$ would be to learn the same policy, I believe, using simpler regularization. The paper should show the result of such a simplification of the method presented in the paper. This simplification would still use reward to regularize $Q_\theta$, but without a use of a probability distribution over rewards, i.e., without the use of a second NN.

The addition of $q_\phi$ is sensible overall if one wants to have a (normal) distribution over rewards for every (s,a). But still, the impact of having $q_\phi$ should be demonstrated empirically as I described in the paragraph above.

There is something wrong about this sentence: "where given the graphical model in figure 2 the reward can be seen as a latent representation of the policy." In RL, the reward is myopic and instantaneous, and the Q-values have to learned to represent a policy. The Q-values take into account long-term consequences of actions, whereas R(s,a) does have this information in MDPs.

The goal of this paper is to learn reward, but on p. 7 the authors said that in their experiments the performance is measured with respect to action matching", and one of the baselines mentioned on the same page can recover the reward function as well. I feel that the rewards extracted by those methods should be compared with the rewards learned by the new method. This is a missing exploration.

Minor problems:

Introduction is of high quality, but this sentence: "This redirection to obtain policies in a manor we shall refer to as apprenticeship learning (AL) introduces its own challenges, particularly in the offline setting." does not explain apprenticeship learning. Please provide a link with IL and IRL that precede this sentence.

One page 4, "they still" -> "there still"

In bibliography "bayes" -> "Bayes" or "gans" -> "GANs"

Summary:

The authors learn the network $Q_\theta$, and they use another network $q_\phi$ to regularize it. It seems that the same empirical results could be achieved without the second network because the first network could be regularized directly. The experiments showing the results of the simpler method would be highly desirable. The link with the autoencoder is not clear because the two networks that are used in this paper are leaned separately and the original inputs are given to the decoder, i.e. the decoder does not decode using a low-dimensional representation. It is not clear why this approach is equated with an autoencoder. The decoder does not decode using the low-dimensional embeddings, and it uses the low dimensional embeddings for regularization only. The low dimensional embeddings are not even given as input to the decoder.

This is interesting work with a good potential, but the above reasons don't allow me to suggest acceptance of the current version of the paper.

------
Added after the discussion period: Thank you for answering my questions and the lively online interaction. You explained a few things to me, and I could see that you understood my point about VAEs. That was a great outcome. I hope that that you will address the  points that we discussed, where showing an honest link with VAEs will be highly desirable for your future readers.

---

> ### Author Response · Authors · 2020-11-16
> **Author response**
>
> Thank you for the review and your thoughtful comments. We have posted a revised version of the paper which we hope address some of the concerns raised, please also see below some specific responses to points raised in your review.
>
> “This must be a coordinate ascent algorithm then” We think there is a little misunderstanding here, and we have revised Section 3 to include a dedicated section on the implementation in order to clarify the details including pseudocode for the algorithm. With that being said we should be clear now that we are not doing coordinate ascent with an alternating scheme of updating $\theta$ and $\phi$ - both are updates with a single evaluation of the loss and as such the error of each network affects the other at every update. You are correct in that we are not doing exactly what an auto-encoder traditionally would, and in Section 3 we are careful to make the point that this is not a VAE although we think it draws a very useful analogy to think about what is going on. In particular while the input is demonstration and the output is a policy - we consider these two things to be equivalent to a degree, since especially in the offline setting a policy is as close as we will get and it naturally induces demonstration (indeed with a bijection between policy and demonstration occupancy measure as shown in [1]). We appreciate though that the path from demonstration to reward and then from reward to policy is not as simple and clean as in a traditional application of the AEVB algorithm of [2] - this is where our contribution lies in being able to perform this kind of inference for the problem at hand. We have modified figure 1 in the paper to better represent the differences to a traditional VAE in order to explain what is happening better.
>
> “I would further claim that $q_\phi$ is not even required…” This is true and we agree with the point, if your only goal is for learning an effective policy $Q_\theta$ then it is not necessary to learn $q_\phi$ and it is possible to get some level of the same effect by just applying a regularisation scheme on the induced reward - this is actually what RCAL [3] does to an extent and so we do actually compare our method against that strategy in the paper. We would like to emphasise thought that the problem here is that one doesn’t end up with a distribution over reward - which is the main point of this contribution. Note that also we present AVRIL as a framework that allows for BIRL at scale and so there is nothing that forces the posterior to be a normal distribution - we just use that as a simple example in the paper. Really you can substitute any parameterised distribution in place, which we emphasise in the fourth paragraph of Section 3.
>
> “In RL the reward is myopic...” While certainly the reward received at each time-step is myopic in RL, the overall reward (function) that is an object in the MDP is not. This reward (function) then dictates an optimal policy given fixed transitions and so in that sense it represents the policy abstractly.
>
> “The goal of this paper is to learn reward…” It should be noted that when ground-truth reward is unavailable, as is the case for the medical data, then measuring performance by action matching is standard in the literature (per [4] for example), as this reflects well the ability of the learnt reward to induce a policy that is close to the demonstrator. In these examples as well AVRIL is really the only method that can reasonably produce a posterior distribution over the reward. In order to visualise better and quantify the learnt reward we have also now included in Section 4 a toy example on a gridworld environment - we demonstrate that we can both recover the reward and also that the epistemic uncertainty produced well reflects the knowledge of the state-space.
> Thank you for your comments on the introduction, we have rephrased the apprenticeship learning sentence to be clearer on how it relates to the other paradigms.
>
> [1] Ho, Jonathan, and Stefano Ermon. "Generative adversarial imitation learning." Advances in neural information processing systems. 2016.
> [2] Kingma, Diederik P., and Max Welling. "Auto-encoding variational bayes." arXiv preprint arXiv:1312.6114 (2013).
> [3] Piot, Bilal, Matthieu Geist, and Olivier Pietquin. "Boosted and reward-regularized classification for apprenticeship learning." Proceedings of the 2014 international conference on Autonomous agents and multi-agent systems. International Foundation for Autonomous Agents and Multiagent Systems, 2014.
> [4] Lee, Donghun, Srivatsan Srinivasan, and Finale Doshi-Velez. "Truly batch apprenticeship learning with deep successor features." Proceedings of the 28th International Joint Conference on Artificial Intelligence. AAAI Press, 2019.

---

> > ### Comment · AnonReviewer4 · 2020-11-23
> > **Reviewer's response**
> >
> > Thank you for your answers. I don't have any other questions, except for one comment.
> >
> > Re: "While certainly the reward received at each time-step is myopic in RL, the overall reward (function) that is an object in the MDP is not. This reward (function) then dictates an optimal policy given fixed transitions and so in that sense it represents the policy abstractly."
> >
> > It looks that what you call "overall reward (function)" is the value function in MDPs and RL. If this is what your encoder returns, then you should clarify in your paper whether your R in Fig. 1 is the instantaneous reward (R(s,a) in MDPs) or the value function (i.e., V(S) or Q(s,a)). This is a big difference. I believe that you could see in your gridworld example whether the output R of the encoder was the reward of the value function. If you are learning the V- or Q-functions, then this should be explained in many places in your paper since you are saying everywhere that you are learning the reward R. In RL, R refers to myopic reward, and V- or Q- to the value function that represents the policy indirectly. Please make it clear in the paper. Or perhaps you are not sure? The values that your encoder is learning can probably be anything between the immediate reward in MDPs and the value function?

---

> > > ### Author Response · Authors · 2020-11-24
> > > **Clarifying reward**
> > >
> > > Thank you very much for your reply! We hope our earlier response appropriately addressed your initial concerns.
> > >
> > > On this final point we feel our difference is simply semantic and nothing fundamental about our view of the structure of the MDP. Firstly, our encoder learns R and not V (Q is learnt by the decoder). We hope this can be seen and is clarified on examining the constraint of equation (10) making use of the Bellman equation - and can be seen empirically in our gridworld example (Figure 5c).
> > >
> > > When we say then that R dictates an optimal policy we mean it in a more indirect way than either V and Q - for example if you fix an MDP (S,A,T,$\gamma$,R) then that implicitly defines an optimal policy $\pi$ (we’re not claiming to immediately know that policy, it would be obtained by forward RL, just that it exists and crucially is a function of R). So even though R defines a myopic reward at each time step, the effect of R everywhere defines the policy.
> > >
> > > We will clarify this in the paper further, thanks again.

---

> > > > ### Comment · AnonReviewer4 · 2020-11-24
> > > > **auto-encoding**
> > > >
> > > > Thank you for your reply. In my last question I wanted to make sure that you use correct terminology in your paper, i.e., you make a proper distinction between the reward and V- or Q-functions.
> > > >
> > > > I am in general fine with your other responses, but I am still not sure if the title of your paper should include the word "auto-encoding". Reviewer 2 raised the same question and you said in your response to rev 2 (which is also what you wrote in your response to me talking about a bijection): --- and in particular that although "the input is demonstration and the output is the policy" on some level especially in the offline setting these are equivalent since the policy (combined with the given environment) would naturally induce demonstration. If you did auto-encoding where the demonstrations were given as input to the encoder, and the policy was at the output of the decoder, then I would agree with your explanation. But, your demonstrations are fed into the decoder directly. Also, in response to Rev. 2 that I quoted above you said that you are auto-encoding policies, but the title of the paper says "Auto-Encoding Reward".
> > > >
> > > > This is a good piece of work (there is no question about that), but there is some lack of clarity with respect to how the term auto-encoding is used. I would argue that the term "auto-encoding" should be removed from the title of the paper. Discussing the connections with VAEs would still be very useful in this paper, but saying that there is auto-encoding here is probably too much.

---

> > > > > ### Author Response · Authors · 2020-11-24
> > > > > **Auto-encoding**
> > > > >
> > > > > Thank you for your reply. On reflection we have removed the term ‘auto-encoding’ from the title given the point of additional input going into the decoder - we very much appreciate your help in positioning this paper in the literature to be as clear as possible. We hope this removes any potential confusion and everything (with respect to R, V, and Q as well) should be correctly reflected now in the updated manuscript we have uploaded.
> > > > >
> > > > > On the point of auto-encoding reward/policy - this was sloppy terminology on our part, since the reward is produced from an auto-encoding of the policy, we have now removed this.
> > > > >
> > > > > Thanks once again for your recommendations.

---

> > > > > > ### Comment · AnonReviewer4 · 2020-11-24
> > > > > > **No more questions from me**
> > > > > >
> > > > > > Thank you for your reply. I can confirm that I don't see any other problems, and I will increase my score. As I said in my previous post, this is a good work overall, and I hope that it will be accepted.

---

> > > > > > > ### Author Response · Authors · 2020-11-24
> > > > > > > **Thanks**
> > > > > > >
> > > > > > > Thank you! We very much appreciate all of your feedback and your very encouraging review!

---

### Official Review · AnonReviewer1 · 2020-10-28

**Rating:** 6
**Confidence:** 3

**Review:**

The paper proposes a Bayesian inverse reinforcement learning algorithm that learns a distribution over reward functions from offline demonstrations. The key idea is to avoid solving an MDP in the inner loop by maximizing the likelihood of the demonstrations with respect to a learned Q function, and to fit a distribution over reward functions to minimize the squared Bellman error with respect to the learned Q function. Experiments on an action prediction task in the healthcare domain as well simulated locomotion and video game tasks suggest that the proposed method is competitive with prior methods for imitation learning.

Overall, I think this paper explores an interesting and impactful idea. There are similarities to the inverse soft Q-learning method [1] for learning the demonstrator's internal dynamics model, which involves simultaneously fitting a Q function to maximize the likelihood of demonstrated actions and to minimize the squared Bellman error with respect to a parametric model of the dynamics. The key difference between ISQL and the proposed method is that the proposed method learns the demonstrator's reward function instead of their internal dynamics model, and that the proposed method learns a distribution over reward functions instead of a point estimate.

While I think this paper is a promising initial step, there are a few issues:

The main issue is that the action prediction experiments show small improvements over the AUC of prior methods on the healthcare dataset, and that AVRIL performs slightly worse than prior methods on several of the control benchmark tasks. One explanation for this relatively poor performance on imitation learning benchmarks could be that standard deep Q-learning does not perform well in the offline RL setting [2], and that AVRIL might share some of these issues because it minimizes the squared Bellman error with respect to learned Q values.

In general, the evaluations could do a much better job of illustrating the benefits of Bayesian IRL vs. standard imitation learning or IRL methods. The current paper evaluates AVRIL's ability to predict actions or imitate demonstrations in the demonstrator's MDP. Instead, one could test AVRIL's ability to learn a reward function that transfers to a new environment with a different initial state distribution or different dynamics -- something that IRL methods can do much better than standard imitation methods. One could also evaluate AVRIL's ability to compute high-confidence bounds on imitation policy performance, as in the evaluations for other Bayesian IRL methods like B-REX [3]. Another possibility is to use AVRIL to learn a distribution over reward functions, then perform risk-averse imitation, as in the experiments for inverse reward design [4]. Similarly, one could potentially use AVRIL as a component in an active learning method for learning rewards, like active inverse reward design [5].

The interpretability analysis on page 8 does not highlight the benefits of learning a distribution over reward functions. The confidence bounds in the top of Figure 4 don't vary substantially, and the bottom of the figure doesn't illustrate AVRIL's uncertainty over the learned weights. It would be helpful to see if, for example, AVRIL learned tight confidence bounds for state-action pairs that are close to those seen in the demonstrations, and wider bounds for out-of-distribution pairs. Furthermore, it would be nice to see if different samples of reward weights from the learned distribution yield distinct, but equally plausible, hypotheses to a human supervisor.

Typos:
 - manor -> manner (paragraphs 2 and 3)
 - In the "Apprenticeship through rewards" paragraph on page 2, "Herman et al. (2016)" should use \citep instead of \citet
 - "it's" -> its (last line on page 5)

1. https://arxiv.org/pdf/1805.08010.pdf
2. https://arxiv.org/pdf/2005.01643.pdf
3. https://arxiv.org/pdf/1912.04472.pdf
4. https://arxiv.org/pdf/1711.02827.pdf
5. https://arxiv.org/pdf/1809.03060.pdf

Update after rebuttal
-----
Thank you to the authors for addressing my concerns. I have updated my score.

---

> ### Author Response · Authors · 2020-11-16
> **Author response**
>
> Thank you for the review and your thoughtful comments. We have posted a revised version of the paper which we hope address some of the concerns raised, please also see below some specific responses to points raised in your review.
>
> “action prediction experiments show small improvements...” It’s worth reiterating that this is a comparison of AVRIL against dedicated imitation learning algorithms - i.e. those whose sole purpose is to produce an imitator policy and which don’t provide any additional insight. To be clear though AVRIL on the other hand is principally a method for obtaining a posterior distribution over the reward, something that no other current method is capable of in these settings, making it much stronger than the other competition in that resect. What our experiments on MIMIC and the control tasks demonstrate then is that we do not lose any predictive power from the policies, and indeed the induced regularisation of the policy can lead to very strong performance while granting us the insight of the reward posterior.
>
> “the evaluations could do a much better job of illustrating the benefits of Bayesian IRL vs. standard imitation learning or IRL methods.” We agree that these are very interesting directions to take BIRL in and as a result of your useful suggestions we have updated Section 2 (namely the paragraph “The usefulness of uncertainty”) to include some of these examples as to reasons why uncertainty is a very important thing to pay attention to when trying to uncover preferences. Our initial pitch highlighted doing Bayesian inference for the sake of Bayesian inference (or at least that max-margin/max-ent was just not sufficient) but this is absolutely a very important aspect that should be emphasised more. WIth respect to developing the experiment section please see the next paragraph.
>
> “The interpretability analysis…” As in the above point we agree that this example does not make the most of this aspect of our algorithm - we have then now added in Section 4 an extra demonstration on a toy gridworld example (where we can very easily visualise everything) the learnt reward and its associated uncertainty. This highlights exactly the point you make about leaning much tighter confidence around states that are in-distribution while being relatively uncertain about states that we just don’t see from the expert demonstrator for whatever reason.
>
> Thank you for pointing out these typos - we have corrected them in the updated manuscript.

---

### Official Review · AnonReviewer2 · 2020-10-31
**A reasonable contributions on Bayesian Inverse RL**

**Rating:** 7
**Confidence:** 4

**Review:**

This paper presents a method, called Auto-encoded Variational Reward Imitation Learning (VARIL). The presented idea is to learn a probabilistic (Gaussian) vector representation of the implicit reward function, so that the policy (Q) function can be trained in scale. Authors demonstrated that the presented methods can outperform (recently proposed) existing methods over several benchmarks in the offline RL.

Strong points
- Literature in inverse/offline RL is well illustrated. Especially, limitations of existing methods (impractical for modern, complicated, and model-free task environments) are clearly explained.
- The proposed framework/algorithm outperform existing state-of-the-art in several benchmarks.
- Contributions and presented models are easy to follow.

Weak points
- The engineering details are not well explained. I agree with authors that an arbitrary function approximator can be used for encoder. However, descriptions as in Figure 1 are too abstract.
- There are some typos (e.g., the the, figure ...)

It would be good to explain more the training procedure in the paper. As an example, it is not sure why the presented encoder is called as auto-encoding reward when the input is demonstration and the output is the policy.

---

> ### Author Response · Authors · 2020-11-16
> **Author response**
>
> Thank you for the review and your thoughtful comments. We have posted a revised version of the paper which we hope address some of the concerns raised, please also see below some specific responses to points raised in your review.
>
> We have updated the paper to reflect more of the details of implementation in Section 3 including pseudo-code for the algorithm. Although we should point out that AVRIL really reflects a framework to allow for scalable BIRL and so a lot of the engineering details (like the implementation of the encoder and decoder) are problem specific. Of course if there are further specific implementation/engineering details we will be happy to provide them.
>
> With respect to our choice of the term “auto-encoding” we try to clarify our position in Section 3 under the paragraph titled “The generative model view”. We try to be clear that this is not a VAE, although it contains similarities to the AEVB algorithm of [1] - and in particular that although “the input is demonstration and the output is the policy” on some level especially in the offline setting these are equivalent since the policy (combined with the given environment) would naturally induce demonstration.
>
> [1] Kingma, Diederik P., and Max Welling. "Auto-encoding variational bayes." arXiv preprint arXiv:1312.6114 (2013).

---

### Official Review · AnonReviewer3 · 2020-11-01
**Potentially great contribution to BIRL, some details missing**

**Rating:** 6
**Confidence:** 3

**Review:**

The authors propose a Bayesian IRL algorithm based on variational inference by learning a policy similar to the one by the demonstrator and inferring the reward function at the same time without sampling. This is achieved by maximizing the ELBO of the posterior of rewards given the demonstrations assuming a softmax probability over Q-values. The contribution is to parameterize the policy and the reward function separately and connecting them through a constraint, which can be reformulated to lead to a single objective.

The paper is well written. Generally, I was particularly impressed by the authors’ explanations of related literature in sections 2 and 3. With one exception. The authors should consider citing Rothkopf & Dimitrakakis, Preference elicitation and inverse reinforcement learning, (2011), particularly, if they cite Ramachandran & Amir, (2007) as well as citing Dimitrakakis & Rothkopf, Bayesian multitask inverse reinforcement learning, (2011) if they cite the hierarchical Bayesian extensions Choi & Kim (2011) and Choi & Kim (2012).

The evaluations are not totally convincing, though. For the final version of the paper, I would encourage the authors to provide simulations on standard IRL problems used in the literature and quantifying performance e.g. with the cumulative loss of the Q-values wrt. the optimal or demonstrator policies. Also, the current manuscript does not provide probabilistic evaluations of the inferred policies or reward functions nor an assessment of how well the variational approximation with GP prior fairs in reward function inference compared to previous algorithms.

Very little details are provided about the implementations, which is a weak point.

Overall, it would be very helpful if the authors can provide some more details about the implementation and provide evaluations that quantify the inferred reward functions in the rebuttal. With the additional information, this could be a very strong paper that advances BIRL.

The authors are encouraged to release the code to foster open science and reproducibility.  Only the publicly accessible repositories used in this research are mentioned but not whether the authors intend to release their code upon acceptance of their manuscript.

Typo: page 3: “offline setting setting”.

---

> ### Author Response · Authors · 2020-11-16
> **Author response**
>
> Thank you for the review and your thoughtful comments. We have posted a revised version of the paper which we hope address some of the concerns raised, please also see below some specific responses to points raised in your review.
>
> “The authors should consider citing…” Thank you for these suggestions - they are most certainly relevant and we have updated the manuscript in Section 2 to reflect these works.
>
> “The evaluations are not totally convincing…” We include the control environments and the return received by the algorithms when deployed live in the environments as very standard in the IRL/IL literature. We should also note that when ground-truth reward is unavailable, as is the case for the medical data, then measuring performance by action matching is standard in the literature (per [1] for example), as it reflects well the ability of the learnt reward to induce a policy that is close to the demonstrator. In these examples as well, AVRIL is really the only method that can reasonably produce a posterior distribution over the reward. In order to visualise better and quantify the learnt reward we have also now included in Section 4 a toy example on a gridworld environment - we demonstrate that we can both recover the reward and also that the epistemic uncertainty produced well reflects the knowledge of the state-space.
>
> “Very little details are provided about the implementations…” We have updated the paper to reflect more of the details of implementation in Section 3 including pseudo-code for the algorithm. Although we should point out that AVRIL really reflects a framework to allow for scalable BIRL and so a lot of the engineering details (like the implementation of the encoder and decoder) are problem specific. It’s our policy that we intend to release full code upon acceptance of the paper which will demonstrate implementations for the example settings we use in the paper. Of course if there are further specific implementation/engineering details we will be happy to provide them.
>
> Thank you for pointing out these typos - we have corrected them in the updated manuscript.
>
> [1] Lee, Donghun, Srivatsan Srinivasan, and Finale Doshi-Velez. "Truly batch apprenticeship learning with deep successor features." Proceedings of the 28th International Joint Conference on Artificial Intelligence. AAAI Press, 2019.

---

### Author Response · Authors · 2020-11-16
**Thank you for your reviews**

We would like to thank all of the reviewers for their time and thoughtful comments and reviews. We have now posted a revised version of the paper which we hope addresses the points raised by everyone, and posted specific responses to individual reviews.

We are eager to hear any further comments or thoughts and so encourage further discussion should anything still be unclear. Thanks in advance.

---

### Author Response · Authors · 2020-11-24
**Title clarification**

We have now uploaded a revised version of the paper. Following helpful discussion and feedback from the reviewers we have decided to remove the term "Auto-encoding" from the title of the paper to clarify the positioning of the paper and we emphasise in the paper the connections and departures from our work to the more traditional use of the term.

Once again thank you to all of the reviewers for their useful suggestions for improving the paper.

---

### Decision · Program_Chairs · 2021-01-07
**Final Decision**

**Decision:**

Accept (Poster)

**Comment:**

The reviewers agree that the submitted paper is of high quality and provides a promising approach/framework for Bayesian IRL. Certain concerns regarding details of the implementation and evaluation have already been addressed by the authors during the rebuttal phase, and also the title of the paper was adjusted in line with discussions with the reviewers. For the final paper, the authors should make sure to clearly highlight the advances of inferring a distribution over rewards (this is already partly done by the added grid world experiments) and discuss relations to VAEs as the initially had in mind and even in the paper title. Beyond that, the should of course also address other reviewers’ comments.